# MOZART, a QSAR Multi-Target Web-Based Tool to Predict Multiple Drug–Enzyme Interactions

**DOI:** 10.3390/molecules28031182

**Published:** 2023-01-25

**Authors:** Riccardo Concu, Maria Natália Dias Soeiro Cordeiro, Martín Pérez-Pérez, Florentino Fdez-Riverola

**Affiliations:** 1LAQV@REQUIMTE/Department of Chemistry and Biochemistry, Faculty of Sciences, University of Porto, 4169-007 Porto, Portugal; 2CINBIO, Department of Computer Science, ESEI—Escuela Superior de Ingeniería Informática, Universidade de Vigo, 32004 Ourense, Spain; 3SING Research Group, Galicia Sur Health Research Institute (IIS Galicia Sur), SERGAS-UVIGO, 36213 Vigo, Spain

**Keywords:** drug–enzyme interaction, QSAR, machine learning, artificial neural network, drug, enzyme, QSAR tool

## Abstract

Developing models able to predict interactions between drugs and enzymes is a primary goal in computational biology since these models may be used for predicting both new active drugs and the interactions between known drugs on untested targets. With the compilation of a large dataset of drug–enzyme pairs (62,524), we recognized a unique opportunity to attempt to build a novel multi-target machine learning (MTML) quantitative structure-activity relationship (QSAR) model for probing interactions among different drugs and enzyme targets. To this end, this paper presents an MTML-QSAR model based on using the features of topological drugs together with the artificial neural network (ANN) multi-layer perceptron (MLP). Validation of the final best model found was carried out by internal cross-validation statistics and other relevant diagnostic statistical parameters. The overall accuracy of the derived model was found to be higher than 96%. Finally, to maximize the diffusion of this model, a public and accessible tool has been developed to allow users to perform their own predictions. The developed web-based tool is public accessible and can be downloaded as free open-source software.

## 1. Introduction

Enzymes are critical components in our lives since they are responsible for catalyzing almost all the chemical reactions in our bodies and cells. Enzymes are primarily proteins and one of their most important features is the high selectivity and specificity against their substrates [1]. For this reason, and the fact that they regulate several fundamental reactions in our body, enzymes are excellent drug targets and are increasingly attracting the attention of scientists involved in the drug development process. In fact, dysregulation of enzymes is involved in severe disease. For instance, the enzyme family 1.1, alcohol dehydrogenase, is involved in breast neoplasms, Alzheimer’s and carcinoma, amongst others [2,3,4]. Aldehyde dehydrogenase, enzyme class (EC) 1.2, is also associated with Alzheimer’s and breast neoplasms [5]. Phosphotransferases, EC 2.7, are involved with head and neck neoplasms, osteoarthritis and stomach neoplasms [6,7,8].

Consequently, an accurate prediction of drug–target interaction is clearly essential.

Computational approaches have demonstrated their robustness in this field. One of the most common approaches is the docking simulations which have proven their ability to reveal binding mechanisms and binding sites [9,10,11,12]. However, this approach has some drawbacks. In fact, one key requisite for docking simulation is the availability of a 3D structure of the enzyme. In addition, these studies may be very time-consuming and testing a large number of candidates could be challenging.

Another reliable in silico approach comprises, for example, quantitative structure-activity relationship (QSAR) modelling techniques, as they are much less complicated and time-demanding. This methodology has been used since 1962 when Hansch published the first QSAR study [13]. This approach may predict different properties such as toxicity, physical properties, drug activity, enzyme function, or even the toxicity or properties of nanoparticles [14,15,16,17,18,19,20,21,22,23] using specific features of the molecules, also called molecular descriptors (MD). QSAR methodology has been used together with docking in the development of new drugs. QSAR has been widely used to predict drug activity against specific enzyme targets [24,25,26,27,28,29,30]. However, the vast majority of these models are not implemented in a web server or a free-to-use software and cannot be easily used. In addition, these models are usually developed to predict a single drug–enzyme interaction. In this context, a great step forward was performed by Min et al. [31] who developed a sequence-based predictor called iEzy-Drug to predict drug–target interactions using 258 MD for drugs plus the pseudo amino acid composition for enzymes. The model was built using a total of 2719 interactive enzyme–drug pairs and 5438 noninteractive enzyme drugs collected from Kyoto of Genes and Genomes (KEGG) [32]. The overall accuracy reported by the authors was 91%. That model was finally integrated into a web-server, which needs the enzyme sequence and the drug Simplified Molecular Input Line Entry Specification (SMILES) code to predict the specific interactions between both. SMILES is a universal and state-of-the-art textual notation to represent the chemical connectivity of chemical species [33,34]. However, some of the enzymes included in the interactive enzymes–drug pairs may belong to the same enzyme class (EC) and, in some cases, are isoforms of the same enzyme which may result in an overfitted model. In addition, this approach can predict only one specific interaction at the same time. Moreover, Bleakley et al. developed a drug–target interaction supervised model to predict unknown drug–target interactions. Reaching an accuracy above 90%, this model was developed to predict only four classes of drug–target interaction networks involving enzymes, ion channels, G protein-coupled receptors (GPCRs) and nuclear receptors in humans [35]. In light of what has been referred to so far, this work developed a multi-target machine learning model (MT-ML) based on the artificial neural network (ANN) multi-layer perceptron (MLP) algorithm to predict the interaction between drugs and EC families. This model is able to simultaneously predict the likely or unlikely interaction of drugs against 23 different enzyme classes.

## 2. Results and Discussion

### 2.1. ANN Multi-Target Model

In order to find the best models and the best ANN topology, a broad set of 350 ANN models were run. Although several models with neurons in the hidden layer between 20 and 70 were developed, the best models were found to have a range of neurons in the hidden layer between 40 and 50. Since no substantial improvement was found with the higher number of neurons, models with more than 50 neurons in the hidden layer were discarded. Against those 350, the 10 best models were selected are reported in Table 1. It is important to remark that each model is trained and tested with different subsets.

For each model, the Statistica software randomly splits the dataset into training (70%) and validation (30%) for each model. If more models can be found using this approach, it suggests that the approach is robust and the models are not overfitted. Moreover, the models with the same topology are not the same model since the weights of each neuron in the hidden layer are not the same. In fact, each time a neural network starts to be trained a random weight is assigned to each neuron and from there the algorithm starts the fit of the function. Amongst the 10 best models evaluated in Table 1, the first model in Table 1 was selected to be integrated into the MOZART online platform, MLP 39-50-2. The topology of the model indicates that the model uses the 39 MD selected with the forward stepwise process, 50 layers in the hidden layer and has 2 outputs, interacting and non-interacting drugs against enzymes.

The model shows an overall accuracy of 96.26% and is able to correctly classify 60,185 pairs out of 62,524. More specifically, the model was able to correctly classify 42,369 out of 43,767 (96.81%) pairs and a total of 17,816 out of 18,757 (94.98%) in the training and validation sets, respectively. These statistics are reported in full in Table 2. In addition, Appendix A also reports for all the cases their respective classification, whether they belong to the training or to the validation set, MD values, CHEMBL ID, and so forth. It is important to remark that each model is trained and tested with different subsets as reported in Appendix A.

The Matthews correlation coefficient (MCC) was also calculated, which, for our best model, was 0.92 [36]. Note that the closer the MCC is to one, the better the classifying ability of the model. A better threshold for evaluating the a priori classification probability can be inferred by means of the ROC curve. Considering this curve describes a relationship between the TPR versus FPR, higher values of the area under the curve show a high performance of the model. As Figure 1 shows, one can rely on the fact that the present MTML-QSAR model is not a random classifier, but instead a truly statistically significant classifier, since the area under the ROC curve is significantly higher (=0.96) than the area under the random classifier curve (=0.5). The curve for good, moderate and worse models was also reported.

Moreover, this model is able to predict the interaction of a specific drug against one or multiple enzyme family targets. To do so, we calculated the accuracy of the model predicting the interaction against the 23 enzyme families included in the model. Table 3 reports the model performance over the family subclasses. Please note that interacting pairs are those pairs where the drug is interacting with the enzyme.

As seen in Table 3, the model was able to achieve a very high rate of accuracy in each subclass, except in the case of the specificity of the EC 7.2 in the inactive cases and 1.11, 1.5 2.5 and 3.2. In any case, for these subsets, the overall accuracy is still very high.

### 2.2. Web-Based Tool

A web-based tool was implemented using the Spring-Boot JAVA framework (https://spring.io/projects/spring-boot) in conjunction with the Bootstrap 4 library (https://getbootstrap.com/) to allow easy access to execute the developed model. This section provides a step-by-step tutorial to guide the user and illustrate how easy it is to use the developed model and obtain predictions about the interactions between drugs and enzymes. Figure 2 resumes the different platform steps to obtain the generated model predictions. These are as follows:

*Step 0. Online test or download and execution*. The developed web-based tool is public and available at http://sing-group.org/mozart. However, the software and source code are also available for their private use as free open-source software. In this sense, to execute the MOZART (coMpOund enZyme interAction pRedicTor) web-based tool on a desktop or server, it is necessary to download and compile the Java code from https://github.com/mpperez3/MOZART or download the runnable java JAR from https://zenodo.org/record/7410843. The user should ensure that Java 8 or higher is installed in their system (run the java-version command to confirm). Finally, the Mozart platform should be executed with the command java -jar MOZART-1.0-SNAPSHOT.jar.

*Step 1.* Use a web browser to access the public version of MOZART platform at http://sing-group.org/mozart or an installed private version at http://localhost:8080. A drop file area will be seen at the top of the web page to upload a file with multiple SMILES and an input box to insert a unique chemical SMILES. To obtain the model predictions, there are two possibilities:

*Step 2A. Perform a batch analysis.* The developed platform allows users to upload a file with multiple chemical SMILES to obtain a prediction for each of them. The uploaded file must meet the following requirements: (*i*) the uploaded file needs to have the .txt or the .tsv extension. (*ii*) The uploaded file must contain one SMILES per line, and (*iii*) it must contain less than 100 SMILES (otherwise, only the first 100 lines will be analyzed). To help the user understand the input file format, there is a dummy example file available to the visitor at http://sing-group.org/mozart/file/exampleSmiles.tsv. The tabular (tab “\t”) separated file, must contain at least two well-identified columns. One column with an “id” and another column “SMILE” with the chemical compound. The first column “id” must contain a free-text descriptor to identify the specified input compound in the final result table (e.g., “has:7173”) and the second column “SMILE” must contain the chemical descriptor of the compound to evaluate.

*Step 2B performs a unique SMILES analysis.* MOZART platform allows users to perform a fast SMILES prediction by inserting the chemical SMILES at the textual input form, similar to web search engines. Once the user has written the text into the textual input, they must press the enter key or the brain button to submit the SMILES and start the analysis.

*Step 3.* Once all SMILES are submitted, the system shows the current state of prediction analysis to help the visitor check the process is running smoothly.

*Step 4.* Once the model predicts the interaction between drugs and EC families, the platform outputs the data in a heatmap table, showing the visitor the model confidence for each family. In the event that the uploaded SMILES was malformed, the platform indicates it to the user, showing an error message in the specific SMILES row. In this sense, the output platform table (or any of the output files) will contain one row for each SMILES uploaded to the platform, one column with the textual identifier specified in the previous step and one column with the predicted confidence interaction (from 0 to 1) of the specific drug against each enzyme family of the model. Figure 2 shows the predicted confidence for the dummy example file. As can be seen, one compound has an incorrect SMILES (red background), which exemplifies the potential of the platform to identify and warn the visitor of incorrect or unprocessable SMILES. On the other hand, three uploaded compounds have a positive predicted interaction against one or multiple EC. These are all columns with a green background and confidence greater than zero. For example, the MOZART model has predicted that the compound “hsa:7173” (with the “CC(CN(CN(C)C)CN1c2ccccccc2CCc2ccccccc21” SMILE) has a positive interaction with the EC family 1.1 and 1.5 with a confidence of 1. To allow the user to save the model results, the web-based tool supports the downloading of the output table in standard file formats such as CSV (Comma-separated values), PDF (portable document format) or XSL (Spreadsheet office format). Furthermore, to navigate among results, the platform allows sorting the resulting table by each family, searching for a specific SMILES, and filtering the visible columns.

## 3. Materials and Methods

### 3.1. Dataset

The initial dataset used in this work was retrieved from the literature [37,38,39] and was updated using the Kyoto Encyclopedia Of Genes and Genomes (KEGG) [40,41,42] and CHEMBL [43] to retrieve all the known drug–enzymes pairs. The final dataset consists of a total of 62,524 entries, of which 27,086 represent enzyme–drug interacting pairs, while 35,438 are non-interacting pairs. The complete list of the drugs is given in Appendix A. All the details for enzymes and drugs used can be found in the KEGG and Chembl databases. A specific data curation process was performed to avoid duplicated entries or incoherent data. By so doing, alternative forms of the same compound and duplicates interacting with the same enzyme sub-class were removed from the dataset. Since this is a multi-target model, the same compound interacting with a different enzyme sub-class should not be removed from the dataset. In any case, the complete dataset is reported in the Appendix A (https://zenodo.org/record/7410843).

### 3.2. Molecular Descriptors

For each drug, hundreds of molecular descriptors were calculated. The SMILES code was used as input for the chemistry development kit (CDK) library [42]. This is a freely available open-source Java library that provides methods for many common computational chemistry tasks. This library is able to calculate different types of MD, such as hybrid, constitutional, topological, electronic and geometrical; however, this work used only topological descriptors since drug activity is strictly related to their physicochemical properties, which can be encoded by this kind of descriptors [44]. In any case, before building the models, a specific feature selection process was performed in order to identify the more relevant MD to be used in the model. In so doing, the forward stepwise procedure carried out enables the selection of an optimal set of thirteen descriptors from an initial pool of more than two hundred fifty. The forward stepwise method employs a combination of the procedures used in the forward entry and backward removal methods. In Step 1, the procedures for forward entry are performed. At any subsequent step where 2 or more effects have been selected for entry into the model, forward entry is performed if possible, and backward removal is performed if possible, until neither procedure can be performed and stepping is terminated. Stepping is also terminated if the maximum number of steps is reached. This procedure is a specific feature of the STATISTICA software. Since this is a multi-target model, it also calculates the mean value for each descriptor and each enzyme subclass and the difference between the value of the MD and the mean value of the enzyme subclasses. These descriptors are reported as <MD> and DMD, respectively, in the Appendix A. The complete dataset with the drugs used and descriptors with their respective values is reported in the Appendix A.

### 3.3. Artificial Neural Network Models

The ANN models were developed using the neural network tool implemented in the software STATISTICA. To develop a model able to predict multiple endpoints using binary classification, the Box–Jenkins moving average was used, which has already been applied in various fields [15,16,27,45,46,47,48,49]. As a result of using a multi-target approach to perform multiple predictions between enzymes and drugs, this model predicts whether a drug may interact with one or more enzyme sub-family targets. To identify the best ANN topology, a broad set of more than 100 models with various topologies were run with a range of 20 to 60 neurons in the hidden layer between. This step, together with the feature selection, is crucial to avoid a problem, albeit unlikely, of overfitting. MLP networks were examined since they usually perform better than other algorithms. The discriminatory power of the model was assessed using the x-fold-validation method, Matthews correlation coefficient and receiver operating characteristics (ROC) curve. This indicator describes a relationship between the model’s sensitivity (the true-positive rate or TPR) versus its specificity (described with respect to the false-positive rate: 1-FPR). The TPR, known as the sensitivity of the model, is the ratio of correct classifications of the “positive” class, while the FPR is the ratio between false positives and all the negative classes. Regarding the cross-validation test, the evaluation was implemented using the STATISTICA software. In so doing, the software in each model automatically splits the entries between training (70%) and validation set (30%). The model is first trained using the training subset and then validated using the validation subset. It is important to highlight that the software randomly assigns entries to train, or validation sets, for each model built. This means that each model is built with a selected number of examples (i.e., training set) and validated with no overlapped selected examples (i.e., validation set). The entries of the validation set are not used while training the model and thus, the validation set could be considered an external test set. Figure 3 depicts the scheme of the training and validation process. As a result, if several models with similar accuracy are built, the overfitted problem is avoided.

## 4. Conclusions

Predicting drug–enzyme interactions is a key step in the development of new drugs and also for drug retargeting. Classical wet methods can be both money- and time-consuming. Due to this, computational approaches should be used in view of the 3R (replacement, reduction and refinement) policy that aims to avoid animal testing as much as possible. This manuscript presents a machine learning multi-target model to predict the interaction of drugs with up to 23 different enzyme sub-classes. The developed model achieved an overall accuracy higher than 96%. This model has been implemented in a web-based tool freely available at http://sing-group.org/mozart and can be downloaded as free open-source software at https://github.com/mpperez3/MOZART or in their java-compiled version at https://zenodo.org/record/7410843. This model with the corresponding web-based tool may represent a great step forward in this field compared with the actual state of the art. In fact, this model has been developed using a very large dataset compared to the published models and is able to make robust and accurate predictions for most of the drug–enzyme pairs. To date, no other models are able to achieve the accuracy of MOZART to multiple predictions.

## Figures and Tables

**Figure 1 molecules-28-01182-f001:**
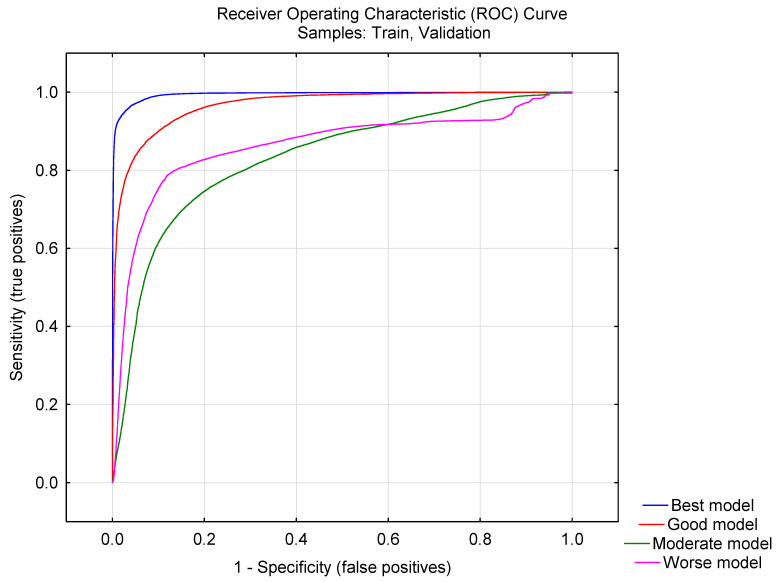
ROC curve for best, good, moderate and worse models found.

**Figure 2 molecules-28-01182-f002:**
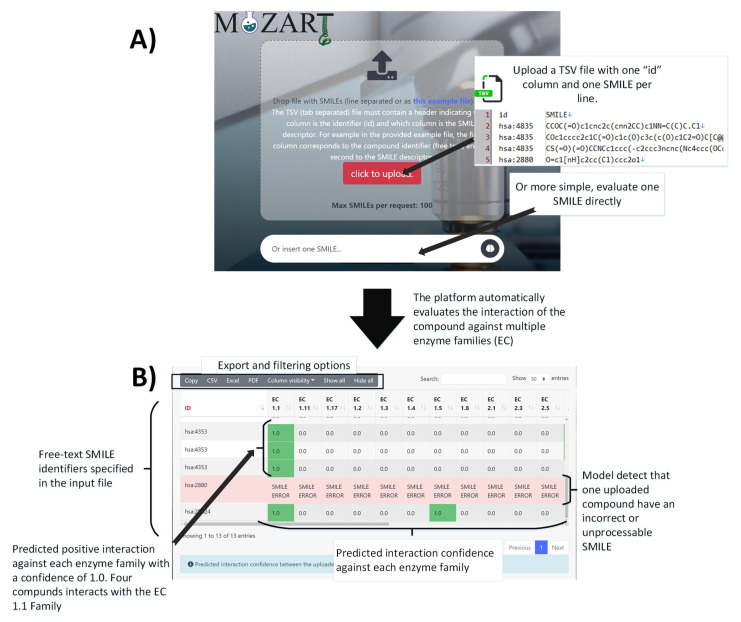
Workflow illustration of the MOZART platform execution. Panel (**A**) depicts the input platform panel, where the user could upload a TSV file or write SMILES in the input box to evaluate their interaction. Panel (**B**) presents the result platform table with the predicted interaction confidence of each SMILES compound against each enzyme family (EC). The background red row depicts an unprocessable SMILES, whereas the green background column depicts each positive predicted interaction against each EC.

**Figure 3 molecules-28-01182-f003:**
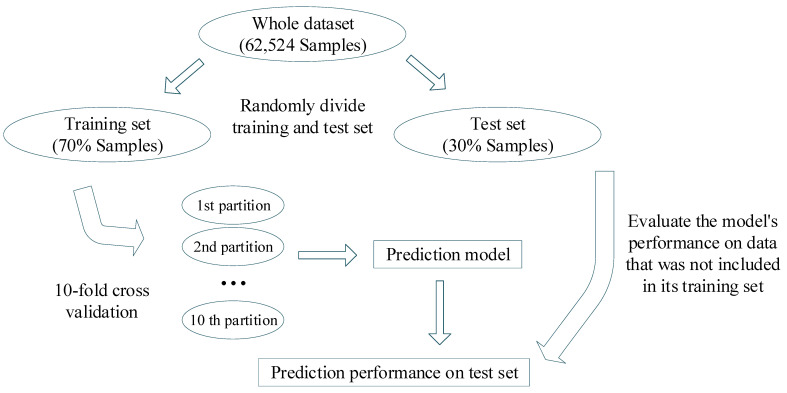
Scheme of the training and validation process.

**Table 1 molecules-28-01182-t001:** The 10 best ANN models.

Model Topology		Inactive *	Active *	Overall
MLP 39-50-2	Total	35,438	27,086	62,524
	Correct	34,247	25,938	60,185
	Incorrect	1191	1148	2339
	Correct (%)	96.64	95.76	96.26
	Incorrect (%)	3.36	4.24	3.74
MLP 39-43-2	Total	35,438	27,086	62,524
	Correct	34,188	25,829	60,017
	Incorrect	1250	1257	2507
	Correct (%)	96.47	95.36	95.99
	Incorrect (%)	3.53	4.64	4.01
MLP 39-50-2	Total	35,438	27,086	62,524
	Correct	34,170	25,888	60,058
	Incorrect	1268	1198	2466
	Correct (%)	96.42	95.58	96.06
	Incorrect (%)	3.58	4.42	3.94
MLP 39-48-2	Total	35,438	27,086	62,524
	Correct	34,157	25,820	59,977
	Incorrect	1281	1266	2547
	Correct (%)	96.39	95.33	95.93
	Incorrect (%)	3.61	4.67	4.07
MLP 39-49-2	Total	35,438	27,086	62,524
	Correct	34,118	25,839	59,957
	Incorrect	1320	1247	2567
	Correct (%)	96.28	95.40	95.89
	Incorrect (%)	3.72	4.60	4.11
MLP 39-41-2	Total	35,438	27,086	62,524
	Correct	34,167	25,839	60,006
	Incorrect	1271	1247	2518
	Correct (%)	96.41	95.40	95.97
	Incorrect (%)	3.59	4.60	4.03
MLP 39-48-2	Total	35,438	27,086	62,524
	Correct	34,242	25,854	60,096
	Incorrect	1196	1232	2428
	Correct (%)	96.63	95.45	96.12
	Incorrect (%)	3.37	4.55	3.88
MLP 39-43-2	Total	35,438	27,086	62,524
	Correct	34,160	25,842	60,002
	Incorrect	1278	1244	2522
	Correct (%)	96.39	95.41	95.97
	Incorrect (%)	3.61	4.59	4.03
MLP 39-49-2	Total	35,438	27,086	62,524
	Correct	34,148	25,762	59,910
	Incorrect	1290	1324	2614
	Correct (%)	96.36	95.11	95.82
	Incorrect (%)	3.64	4.89	4.18
MLP 39-41-2	Total	35,438	27,086	62,524
	Correct	34,167	25,839	60,006
	Incorrect	1271	1247	2518
	Correct (%)	96.41	95.40	95.97
	Incorrect (%)	3.59	4.60	4.03

* Inactive, drugs inactive against enzymes; Active, drugs active against enzymes.

**Table 2 molecules-28-01182-t002:** Statistics for the best ANN model.

Overall			
	Sensitivity	Specificity	Overall
Total	35,656	28,168	63,824
Correct	34,438	26,907	61,345
Incorrect	1218	1261	2479
Correct (%)	96.58	95.52	96.12
Incorrect (%)	3.42	4.48	3.88
Training			
Total	25,339	19,338	44,677
Correct	24,538	18,537	43,075
Incorrect	801	801	1602
Correct (%)	96.84	95.86	96.41
Incorrect (%)	3.16	4.14	3.59
Validation			
Total	10,317	8830	19,147
Correct	9900	8370	18,270
Incorrect	417	460	877
Correct (%)	95.96	94.79	95.42
Incorrect (%)	4.04	5.21	4.58

**Table 3 molecules-28-01182-t003:** Accuracy of the model for each subclass.

EC	Enzyme Subclass Name	Total Entries	Interacting Pairs %	No Interacting Pairs %
1.1	Acting on the CH-OH group of donors	2917	0.944289694	0.996090696
1.11	Acting on a peroxide as an acceptor	887	0.819548872	0.966843501
1.17	Acting on CH or CH2 groups	400	0.964912281	0.962099125
1.2	Acting on the aldehyde or oxo group of donors	27,517	0.989814307	0.82320442
1.3	Acting on the CH-CH group of donors	794	0.985765125	0.990253411
1.4	Acting on the CH-NH2 group of donors	2222	0.981687014	0.938095238
1.5	Acting on the CH-NH group of donors	1291	0.833333333	0.999210734
1.8	Acting on a sulfur group of donors	157	0.914634146	0.866666667
2.1	Transferring one-carbon groups	633	0.983451537	0.980952381
2.3	Acyltransferases	328	0.962962963	0.991902834
2.5	Transferring alkyl or aryl groups, other than methyl groups	183	0.842105263	1
2.6	Transferring nitrogenous groups	67	1	0.8
2.7	Transferring phosphorus-containing groups	2036	0.982942431	0.989981785
3.1	Acting on ester bonds	3639	0.979591837	0.999435188
3.2	Glycosylases	12,598	0.82436189	0.928249045
3.3	Acting on ether bonds	602	0.935897436	0.992366412
3.4	Acting on peptide bonds (peptidases)	723	0.918367347	0.992
3.5	Acting on carbon-nitrogen bonds, other than peptide bonds	58	0.6875	0.976190476
4.2	Carbon-oxygen lyases	3678	0.897035881	0.985841291
4.6	Phosphorus-oxygen lyases	105	0.966666667	1
5.3	Intramolecular isomerases	120	0.983870968	0.965517241
5.6	Isomerases altering the macromolecular conformation	1530	0.961145194	0.986551393
7.2	Catalysing the translocation of inorganic cations	283	0.986013986	0.55

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
