# Peer review of "MOZART, a QSAR Multi-Target Web-Based Tool to Predict Multiple Drug–Enzyme Interactions"

_molecules, 2023, doi:10.3390/molecules28031182_

Round 1

Reviewer 1 Report

In this work, R. Concu et al. reported the novel machine learning multi-target model for the prediction of binding between possible drug candidates and 23 different enzyme classes. The authors performed a relatively extensive and thorough study using a large benchmark set of compounds and tested 100 different ANN models. The study is interesting and described software can be a very useful tool for drug design. However, before publication, the work needs some refinement and corrections.

Major comments:

1.    In the manuscript I found 3(!) different numbers for the size of the dataset. Namely, 66,829 compounds in the abstract (p.1, line 14), 65,115 in Section 2.1 (p.3, line 82), and 63,824 in Section 3.1, (p.4, line 131). The supplementary table also contains 63,824 compounds. Which number is correct? How was the dataset prepared? Page 3, lines 84-85: “A specific data curation process has been performed to avoid duplicated entries or incoherent data”. However, I found in the supplementary table 3,932 duplicates. Also, this dataset includes alternative forms of one compound. For instance, CHEMBL2 and CHEMBL1558, prazosin and prazosin hydrochloride, respectively; CHEMBL611 and CHEMBL125665, terazosin and terazosin hydrochloride, respectively; etc. In other words, these are also duplicate entries. As far as I can tell, an attempt was made by the authors to remove counterions for such compounds, since the supplementary table contains SMILES codes of this kind “CCCC(CCC)C(=O)[O-].[+]”. This observation is relevant for 70 compounds. 977 compounds contain counterions which means they should be controlled so that they are not the alternative forms of one compound. Thus, a quick analysis showed that the size of the dataset could be reduced by almost 10%. I strongly encourage the authors to carefully analyse and prepare the initial dataset as this is critical to the present study.

2.    The description of the used methods must be improved. For example, the description of the calculation and selection of the molecular descriptors is insufficient and unclear. In Section 2.3, is mentioned that 39 descriptors from 250 were selected using “a specific feature selection process” (p. 3, line 102). Does the number “250” is “hundreds of molecular descriptors” mentioned in Section 2.2 (p.3, line 90)? What exactly was “a specific feature selection process”? Why only topological descriptors were used?

3.    The authors first describe the best model and its merits and then describe the top ten models found. In my opinion, the description of the results should be changed. This order of description looks more logical: 100 models were tested, the top 10 were selected, and the best model was determined. Are models „MLP: 39-42:2“ (p.4, line 128) and “MLP 39-42-2” (Table 3, p.8) the same model? If this is correct, why do the data presented for these models in table 1 and table 3 not match? Also, the authors need to reconsider the approach to the design of tables, since, at the moment, most of the tables are not entirely clear and confusing.

4.    The detailed legend should be added to the supplementary table. What does mean the angle brackets in the notation of the descriptor? For example, SPC-5 and <SPC-5>? It can be assumed that this is a descriptor related to an enzyme since its value varies according to the enzyme class. If this assumption is correct, how were the enzyme descriptors calculated? Were descriptors calculated for all enzymes in the enzyme class or just one enzyme? How, in this case, was the enzyme selected (each enzyme class includes a large number of different enzymes)? This information should be added to the main text.

5.    I strongly encourage authors to carefully check all references and their styles. At the moment the references are formatted incorrectly starting from section 2 and do not match the text. For instance, the publication about enzyme role in human gastric cancer (Ref. 4) cannot be used as a reference for the chemistry development kit.

General comments:

1.     The abbreviation and full name should be mentioned the first time they appear in the text. P.1, line 35 “…ec class…”, p.2, line 66 “enzyme class (EC)”. Also, the phrase “ec class” is not correct since the abbreviation “EC” includes the word “class”. The phrase “EC families” also is not correct. Also, note that the same spelling format for the same abbreviation is used throughout the manuscript (this is relevant for all abbreviations in the manuscript, including the names of databases and programs).

2.     P. 1, line 40. Reference should be added to the first sentence.

3.     P.1-2, lines 42-43. In the absence of an available 3D enzyme structure, a 3D structure obtained by homology modelling can be used for docking. Nowadays the different molecular docking programs allow for performing high-throughput virtual screening of millions of compounds within several days.

4.     In my opinion, the statement that the QSAR method was first developed and used in 1976 is not entirely true. According to Cherkasov et al. (DOI: 10.1021/jm4004285), the QSAR modelling was founded by Corwin Hansch in 1962 (https://doi.org/10.1038/194178b0). In addition, the first references to a possible relationship between structure and property were published in 1869 by chemist Crum-Brown: “…There cannot be any reasonable objection against a fact that a relationship between the physiological effect of a compound and its chemical constitution exists….” (PMID: 17230757). I would also like to draw the attention of the authors to the fact that it is impossible to state that the QSAR has a definite advantage over docking (p. 2, lines 46 – 52). Both techniques are widely used in modern drug design and, depending on the task, have their advantages and disadvantages. There are also examples where using a combination of these techniques might be a more efficient approach for the development of new drugs (DOI: 10.1080/07391102.2020.1850355; DOI:10.3390/ijms130911210; https://doi.org/10.1186/s42269-022-00874-1).

5.     What does mean “viable in silico approaches” (p.2, line 46)?

6.     P.2, line 72 – 75 It seems that this information is left from the template and is not relevant to the article.

7.     I recommend to the authors add the names of studied enzyme classes in Table 2.

8.     P. 7, lines 155 – 157 Please make sure that the information in this sentence is correct. Currently, it seems that the data is mixed up.

9.     P. 7, line 164 Please add a reference

10.  P.9, lines 183 – 184 Unfortunately, the link (http://localhost:8080) to the developed software is not working.

11.  The quality of Figure 2 and Figure 3 must be improved.

Author Response

screening of
millions of compounds within several days.
Dear reviewer,
Sentence has been fixed as:” However, this approach has two mainsome drawbacks. On the one
handIn fact, one key requisite for docking simulation is that a 3D structure of the enzyme should
be available. On the other handIn addition, these studies may be very time consuming and testing
a large number of candidates could be challenging.”
COMMENT 4. In my opinion, the statement that the QSAR method was first developed
and used in 1976 is not entirely true. According to Cherkasov et al. (DOI:
10.1021/jm4004285), the QSAR modelling was founded by Corwin Hansch in 1962
(https://doi.org/10.1038/194178b0). In addition, the first references to a possible relationship
between structure and property were published in 1869 by chemist Crum-Brown: “…There
cannot be any reasonable objection against a fact that a relationship between the
physiological effect of a compound and its chemical constitution exists….” (PMID:
17230757). I would also like to draw the attention of the authors to the fact that it is
impossible to state that the QSAR has a definite advantage over docking (p. 2, lines 46 – 52).
Both techniques are widely used in modern drug design and, depending on the task, have
their advantages and disadvantages. There are also examples where using a combination of
these techniques might be a more efficient approach for the development of new drugs
(DOI: 10.1080/07391102.2020.1850355; DOI:10.3390/ijms130911210;
https://doi.org/10.1186/s42269-
022-00874-1).
Dear reviewer,
We agree on that the first QSAR paper was published by Hansch in 1962 and we have changed
the reference. On the other hand, we also agree that both techniques have some advantages as
well as disadvantages and can be used together. In fact, our group in the past published several
papers combining these two techniques, We have changed the paragraph:” Another viable reliable
in silico approaches, comprises for example quantitative structure-activity relationship (QSAR)
modeling techniques, because these are much less complicated and time demanding. This
methodology has been used since 19761962, when Cramer et al.Hansch published the first QSAR
study [13][12]. This approach may predict different properties such as toxicity, physical
properties, drug activity, enzyme function or even predict nanoparticles toxicity or properties[14-
23][13-22] using specific features of the molecules, the so called molecular descriptors (MD).
QSAR methodology has been used together with docking in the development of new drugs
COMMENT 5. What does mean “viable in silico approaches” (p.2, line 46)?
Dear reviewer,
phrase has been changed to ” Another reliable in silico approache”
COMMENT 6. P.2, line 72 – 75 It seems that this information is left from the template and
is not relevant to the article.
The authors are grateful to the reviewer for the word notice. The authors revised the manuscript
to remove all fragments of the template in the new version of the manuscript. Please refer to the
revised version of the results section to evaluate such amendments.
COMMENT 7. I recommend to the authors add the names of studied enzyme classes in
Table 2.
Dear reviewer,
We have added a column with the names of the encyme classes studied
COMMENT 8. P. 7, lines 155 – 157 Please make sure that the information in this sentence
is correct.Currently, it seems that the data is mixed up.
Dear reviewer,
Data has been checked and now should be correct.
COMMENT 9. P. 7, line 164 Please add a reference
Dear reviewer,
A reference has been added to the text.
COMMENT 10. P.9, lines 183 – 184 Unfortunately, the link (http://localhost:8080) to the
developed software is not working.
The authors thank the reviewer for the relevant point raised in this comment. However, the authors
believe that there may have been a misunderstanding on this point. To test the tool, each user will
have to install the program as it is a stand-alone web application and therefore private to each
user. The purpose is that each user installs this free program on their personal computer or server.
Therefore, each URL provided in the article pointed to the internal location on the user's machine
(http://localhost:8080), and access to those resources required the user to install the application
and complete the tutorial in order to complete the access.
Having said that, the authors have addressed this relevant poin by providing a public place on the
Internet (http://sing-group.org/mozart) where anyone interested can test the developed tool
without the need to install it. Please, refer to the new version of the manuscript to evaluate this
amendment.
COMMENT 11. The quality of Figure 2 and Figure 3 must be improved.
The authors are grateful to the reviewer for this constructive suggestion. Therefore, the new
version of the manuscript includes an improved version of both figures.

Reviewer 2 Report

Dear authors,

The article is quite interesting, as it is the validation of a websoftware, which can be of great importance. Mainly because of its high significance in the area.

However, English is very weak. And the text contains a fragment of the template (page1, line 72-75). Figure 2 I consider of little importance for the article, since it shows the best fit, they could put some fits. Even the bad ones could make an A-D figure (best, good-moderate-worse) for us to see.

Author Response

The article is quite interesting, as it is the validation of a websoftware, which can be of great
importance. Mainly because of its high significance in the area. However, English isery
weak.
AUTHORS’ GENERAL RESPONSE: The authors are sincerely grateful to the reviewer for the
positive criticism of the proposed approach. Regarding the appropriate suggestion made by the
reviewer, the presented work was completely revised in the new version of the manuscript to
improve the English.
COMMENT 1.And the text contains a fragment of the template (page1, line 72-75).
The authors are grateful to the reviewer for the word notice. The authors revised the manuscript
to remove all fragments of the template in the new version of the manuscript. Please refer to the
revised version of the results section to evaluate such amendments.
COMMENT 2. Figure 2 I consider of little importance for the article, since it shows the best
fit, they couldput some fits. Even the bad ones could make an A-D figure (best, goodmoderate-
worse) for us to see.
Dear reviewer,
We have prepared a new Figure 2 comparing the results for best, good, moderate and worse model

Reviewer 3 Report

In this study titled 'eMOZART, a QSAR Multi-Target Web Based Tool to Predict Multiple Drug-Enzyme Interactions', the authors Concu et al designed and built a web-based tool eMOZART, a multi-target machine learning (MTML) quantitative structure-activity relationship (QSAR)  model for probing interactions among different drugs and enzyme targets. The methods are well designed and well presented, and the eMOZART tool could make important contributions to the drug design against enzyme targets. Some minor revisions are recommended. 

1. The authors should compare the performance of eMOZART with similar software available and highlight the advantages of eMOZART. 

2. Fig2., more detailed figure legends with better explanation and description of the figure should be provided. 

3. Table 2, the authors should provide the name of each enzyme family as an additional column. 

4. Table 2, the authors should provide more explanation to the table. e.g. what are interacting pairs and no interacting pairs? What score is considered good accuracy? 

5. The dummy tsv file is not accessible. 

6. The link to MOZART website is not provided. The reviewers are unable to evaluate the performance of the website. 

7. It would be helpful if the authors could explain the content of the input (.tsv) file, so that other users could make their own input files.

8. I like the way the authors provided the tutorial. It would be more helpful if the authors can provide more explanation of the output file. 

9. Fig.3 figure legend is too short. Description of the two panels are needed. 

10. Fig.3, why are there so many SMILE errors in the bottom panel? 

11. The are still many typos and grammar errors in the text. The authors should check carefully. Some examples: 

 line 90; Line 108 thasn; Line 158;

Author Response

In this study titled 'eMOZART, a QSAR Multi-Target Web Based Tool to Predict Multiple
Drug-Enzyme Interactions', the authors Concu et al designed and built a web-based tool
eMOZART, a multi-target machine learning (MTML) quantitative structure-activity
relationship (QSAR) model for probing interactions among different drugs and enzyme
targets. The methods are well designed and well presented, and the eMOZART tool could
make important contributions to the drug design against enzyme targets. Some minor
revisions are recommended.
AUTHORS’ GENERAL RESPONSE: The authors are sincerely grateful to the reviewer for the
comprehensive revision of the manuscript and the constructive criticism. Below, a detailed
response is provided to the reviewer’s comments, enumerating the changes that the authors
included in the revised version of the manuscript.
COMMENT 1. The authors should compare the performance of eMOZART with similar
software available and highlight the advantages of eMOZART.
Dear reviewer,
We have introduced in the manuscript new lines in the conclusions section to highlight the
advantages of eMOZART.
“This model with the corresponded corresponding web-based tool may represent a great stepforward
in this field com-pared with the actual state of the art. In fact, this model has been
developed using a very large dataset.compared to the published models and is able to make robust
and accurated predictions for most of the drug-enzyme pairs. No other models are able to achieve
accuracy of eMOZART to multiple prediction so far.”
COMMENT 2. Fig2., more detailed figure legends with better explanation and description
of the figure should be provided.
Dear reviewer,
We have added this sentence in the materials and methods section to describe the ROC curve:
“This indicator describes a relationship between the model’s sensitivity (the true-positive rate or
TPR) versus it’s specifici-ty (described with respect to the false-positive rate: 1-FPR) The TPR,
known as the sensitivity of the model, is the ratio of correct classifications of the “positive” class
while the FPR is the ratio between false positives and all the negative classes”
COMMENT 3. Table 2, the authors should provide the name of each enzyme family as an
additional column.
Dear reviewer,
A column with the name of the enzymes families has been added
COMMENT 4. Table 2, the authors should provide more explanation to the table. e.g. what
are interacting pairs and no interacting pairs? What score is considered good accuracy?
Dear reviewer,
We have add this sentence in the manuscript to clarify the meaning of interacting pair:” Please
note that interacting pairs are those pairs where the drug is interacting with the enzyme.”
In addition, in this contest we can consider a good accuracy above 80% .
COMMENT 5. The dummy tsv file is not accessible.
The authors thank the reviewer for the word notice. However, the authors believe that there may
have been a misunderstanding at this point. The software provided has been developed as a standalone
online tool that each user must install to test the tool and download the dummy files. This
free software is intended for each user to install on their own computer or server. Therefore all
URLs provided in the paper referred to the internal location of the resources (on the own
computer) and they are only accessible if the user installs the programme and follows the tutorial.
Having said that, the authors have addressed this relevant point and the authors have decided to
provide a public online location (http://sing-group.org/mozart) where anyone interested can test
the developed tool without the need to install it, keeping also the option to download and install
it as free open-source software. Please, refer to the new version of the manuscript to evaluate this
amendment.
COMMENT 6. The link to MOZART website is not provided. The reviewers are unable to
evaluate the performance of the website.
As in the previous COMMENT 5 and with the goal of clarifying this situation, the authors have
decided to provide a public online location (http://sing-group.org/mozart) where anyone
interested can test the developed tool without the need to install it. Please, refer to the new version
of the manuscript to evaluate this amendment.
COMMENT 7. It would be helpful if the authors could explain the content of the input (.tsv)
file, so that other users could make their own input files.
The authors appreciate the interesting point raised by the reviewer and agree with the idea that
the content of the input (.tsv) file should be explained. In this regard, the authors have added a
legend to the online application describing how this file can be constructed. Besides, the new
version of the manuscript includes additional information that tries to clarify this relevant point.
Please, evaluate the fragment below:
“Step 2A. Perform a batch analysis. The developed platform allows users to upload a file
with multiple chemical SMILEs to obtain a prediction for each of them. The uploaded file
must meet the following requirements: (i) the uploaded file needs to have the .txt or the
.tsv extension. (ii) the uploaded file must contain one SMILE per line, and (iii) it must
contain less than 100 SMILEs (otherwise, only the first 100 lines will be analyzed). To
help the user understand the input file format, there is a dummy example file available to
the visitor at http://sing-group.org/mozart/file/exampleSmiles.tsv. The tabu-lar (tab “\t”
) separated file, must contain almost two well-identified columns. One column with an
“id” and another column “SMILE” with the chemical compound. The first column “id”
must contain a free-text descriptor to identify the specified input compound in the final
result table (e.g. “has:7173”) and the second column “SMILE” must contain the
chemical descriptor of the compound to evaluate.” Lines 185-193
COMMENT 8. I like the way the authors provided the tutorial. It would be more helpful if
the authors can provide more explanation of the output file.
One more time, the authors agree with the constructive comment of the reviewer. Consequently,
in agreement with the reviewer's suggestions, the new version of the manuscript and the online
platform provide additional information about the output of the tool. Please, evaluate the fragment
below:
“Step 4. Once the model predicts the interaction between drugs and EC families, the
platform output the data in a heatmap table, showing the visitor the model confidence for
each family. In case the upload SMILE was malformed, the platform indi-cates it to the
user showing an error message in the specific SMILE row. In this sense, the output
platform table (or any of the output files) will contain one row for each SMILE uploaded
to the platform, one column with the textual identifier specified in the previous step and
one column with the predicted confidence interaction (from 0 to 1) of the specific drug
against each enzyme families of the model. In this sense, Figure 3 shows the predicted
confidence for the dummy exam-ple file. As can be seen, one compound has an incorrect
SMILE (red background), this exemplifies the potential of the platform to identify and
warn the visitor of incorrect or unprocessable SMILEs. On the other hand, three
uploaded com-pounds have a positive predicted interaction against one or multiple EC.
These are all columns with a green background and confidence greater than zero. For
example, the MOZART model has predicted that the compound "hsa:7173" (with the
"CC(CN(CN(C)C)CN1c2ccccccc2CCc2ccccccc21" SMILE ) has a positive interaction
with the EC family 1.1 and 1.5 with a confidence of 1. To allow the user to save the model
results, the web-based tool supports the downloading of the output table in standard file
formats such as CSV (Comma-separated values), PDF (portable document format), or
XSL (Spreadsheet office format). Furthermore, to navigate among results, the platform
allows sorting the resulting table by each family, searching for a specific SMILE, and
filtering the visible columns.” Lines 199-214
COMMENT 9. Fig.3 figure legend is too short. Description of the two panels are needed.
The authors are grateful to the reviewer for the word of notice and the constructive criticism about
Figure 3. Therefore, in agreement with the reviewer's suggestion, the caption of figure 3 has been
reworded to add more information. Please, refer to the new figure 3 and their legend to evaluate
this new contribution.
COMMENT 10. Fig.3, why are there so many SMILE errors in the bottom panel?
The authors appreciate the interesting point raised by the reviewer and agree that Figure 3 could
lead to confusion. The objective was to demonstrate by using an image that the system is capable
of assessing malformed SMILEs and notifying the user of this event. Therefore, the current
version of the manuscript now includes an updated version of Figure 3. Please, refer to the new
version of the figure to evaluate the changes. Please, evaluate the fragment below:
“ As can be seen, one compound has an incorrect SMILE (red background), this exemplifies the
potential of the platform to identify and warn the visitor of incorrect or unprocessable SMILEs.”
Lines 205-206
COMMENT 11. The are still many typos and grammar errors in the text. The authors
should check carefully. Some examples:line 90; Line 108 thasn; Line 158;
The authors are grateful to the reviewer for the word notice. Regarding the appropriate suggestion
made by the reviewer, the presented work was completely revised in the new version of the
manuscript to improve the English and resolve the typos (please, see the attached certificate).

Round 2

Reviewer 2 Report

Dear authors the improvement of the article was sensible. Since that I indicate the publication of this manuscript 

Reviewer 3 Report

I appreciate the authors' hard work revising the manuscript. All my concerns are addressed. I recommend accept in present form.